# Chemical Synthesis of *Trans* 8-Methyl-6-Nonenoyl-CoA and Functional Expression Unravel Capsaicin Synthase Activity Encoded by the *Pun1* Locus

**DOI:** 10.3390/molecules27206878

**Published:** 2022-10-13

**Authors:** Raika Milde, Arianne Schnabel, Toni Ditfe, Wolfgang Hoehenwarter, Carsten Proksch, Bernhard Westermann, Thomas Vogt

**Affiliations:** 1Department of Cell and Metabolic Biology, Leibniz Institute of Plant Biochemistry, Weinberg 3, D-06120 Halle (Saale), Germany; 2Department of Bioorganic Chemistry, Leibniz Institute of Plant Biochemistry, Weinberg 3, D-06120 Halle (Saale), Germany; 3Department of Biochemistry of Plant Interactions, Leibniz Institute of Plant Biochemistry, Weinberg 3, D-06120 Halle (Saale), Germany

**Keywords:** capsaicin, *Capsicum*, *Pun1*, acyltransferase, BAHD, CoA-ester, enzyme activity, substrate specificity

## Abstract

Capsaicin, produced by diverse *Capsicum* species, is among the world’s most popular spices and of considerable pharmaceutical relevance. Although the capsaicinoid biosynthetic pathway has been investigated for decades, several biosynthetic steps have remained partly hypothetical. Genetic evidence suggested that the decisive capsaicin synthase is encoded by the *Pun1* locus. Yet, the genetic evidence of the *Pun1* locus was never corroborated by functionally active capsaicin synthase that presumably catalyzes an amide bond formation between *trans* 8-methyl-6-nonenoyl-CoA derived from branched-chain amino acid biosynthesis and vanilloylamine derived from the phenylpropanoid pathway. In this report, we demonstrate the enzymatic activity of a recombinant capsaicin synthase encoded by *Pun1*, functionally expressed in *Escherichia coli*, and provide information on its substrate specificity and catalytic properties. Recombinant capsaicin synthase is specific for selected aliphatic CoA-esters and highly specific for vanilloylamine. Partly purified from *E. coli*, the recombinant active enzyme is a monomeric protein of 51 kDa that is independent of additional co-factors or associated proteins, as previously proposed. These data can now be used to design capsaicin synthase variants with different properties and alternative substrate preferences.

## 1. Introduction

Capsaicin and related capsaicinoids are exclusively produced by chili peppers (*Capsicum spec.*) originating from subtropical Central America and have spread since their discovery all over the world [1]. They are synthesized in placental cells and reportedly accumulate in the glandular areas of interlocular septa within ripening fruits [2]. Capsaicin and capsaicinoids elicit pungency by binding to certain ion channels of the transient receptor potential vanilloid family (TRPV1) [3,4]. The binding of capsaicin to the receptor results in an influx of sodium and calcium ions into nociceptive neurons, depolarizing these cells and resulting in the perception of pungency. While numerous reports have been published on the potential effects of capsaicinoids, progress toward the elucidation of the complete biosynthesis of these amides is slow [5,6,7]. Molecular and genetic evidence support the role of the pungency 1 (*Pun1*) locus as the decisive step in capsaicin biosynthesis [8]. The *Pun1* locus contains a BAHD-type acyltransferase, capsaicin synthase (termed AT3), that in principle is capable of catalyzing the required amide bond formation between *trans* 8-methyl-6-nonenoyl-CoA derived from branched-chain amino acid biosynthesis and vanilloylamine derived from the phenylpropanoid pathway, resulting in the formation of capsaicin (Figure 1).

This early genetic evidence for this type of reaction was supported by a combination of genomic and transcriptomic data from chili peppers [9,10]. In both cases, a clear correlation of pungency was established to the reported placental expression of AT3 and several other capsaicin biosynthetic genes. Virus-induced gene silencing of *Pun1* resulted in a severe reduction of capsaicin formation [8], while antibodies that were raised against a heterologously expressed AT3 protein were able to reduce capsaicinoid accumulation in isolated protoplasts [11]. Unfortunately, the corresponding, AT3-like acyltransferase purified in this study did not show any enzymatic activity in vitro, and it remains to be demonstrated that this sequence encodes a functional enzyme. The authors concluded that unknown cofactors, posttranslational modifications, specific oligomerization states, or the cellular environment in the *Capsicum* placenta may be required for the enzymatic activity [11]. A US and Chinese patent (US9951358B2) claimed the successful production of capsaicin by combinatorial cloning and natural fermentation of a CoA-ligase, an amino-transferase (pAMT), and capsaicin synthase encoded by the *Pun1* locus in *Escherichia coli* [12]. In a large screen of engineered *Saccharomyces cerevisiae* using a combinatorial approach of different acyltransferases and CoA ligases, capsaicin could be produced in a bioreactor among several other capsaicinoids [13]. The study was optimized for high nonivamide (nonanoyl-vanilloylamide) production, a compound that was used as a synthetic capsaicinoid analogue in numerous applications. That study elegantly demonstrated that tyramine-*N*-hydroxycinnamoyltransferase from *C. annuum* even outcompeted AT3 and may open a promising alternative for the production of these therapeutically relevant metabolites in a bioreactor. Nevertheless, the determination of capsaicin synthase activity in vitro is still elusive and basic enzymatic properties have never been addressed. 

In this report, we demonstrate the successful functional expression of recombinant capsaicin synthase in *E. coli*. To characterize the enzymatic activity, we synthesized two relevant CoA-precursors and provided data on the substrate specificity as well as preliminary kinetic data of a partially purified, enzyme preparation. The experiments are consistent with a monomeric capsaicin synthase able to catalyze capsaicin formation from *trans* 8-methyl-6-nonenoyl-CoA and vanilloylamine without any co-factor requirements.

## 2. Results

### 2.1. Detection, Cloning, and Functional Expression of Capsaicin Synthase

For the recombinant enzyme production, we selected the published sequence from *Capsicum chinense* [7] and codon-optimized this sequence for its expression in *E. coli*. Among several *E. coli* strains, IPTG-inducible SoluBL21 cells were chosen and proposed for the expression of problematic enzymes, specifically BAHDs [14]. Lowering the growth temperature from the usual 37 °C to 25 °C resulted in a small amount of product formation already in a crude extract. The identification of the product as capsaicin was based on an LC-ESI/MS performed in positive ionization mode. The signal was identical in terms of retention time, molecular mass *m/z* 306.1 [M + H]^+^, and retention time with a commercially available capsaicin standard (Figure 1a and Appendix A). In the crude extracts, we only measured trace amounts of active protein (Figure 2a) that could not be detected by an anti-His-tag antibody. The identification of a protein band on the SDS-PAGE that corresponded to the recombinant capsaicin synthase required additional purification steps to select a single protein band that was consistent with the activity profile. After affinity chromatography of the His-tagged protein and subsequent concentration of the extract, the fraction containing recombinant capsaicin synthase activity was further purified by size exclusion chromatography (SEC) (Figure 2b). Once more, this did not yield pure protein but resulted in a distinct signal that could be detected by the anti-His antibody among a mix of three candidate bands of a similar mass between 45 kDa and 52 kDa (Figure 2c,d). This protein fraction produced capsaicin from *trans* 8-methyl-6-nonenoyl-CoA and vanilloylamine and was used for subsequent functional characterization (Figure 2c,d). At this stage, it was confirmed that the enzymatic activity was consistent with a molecular mass of a monomeric enzyme of around 50 kDa. In order to identify the correct enzyme among the three dominant bands of 52 kDa, 50 kDa, and 48 kDa (Figure 2c), of which only the top one showed a signal with the anti-His-tag antibody, all of the bands were eluted from the gel, subjected to trypsin digestion, and the peptides were subsequently analyzed by LC-MS. 

Among the digested and sequenced protein bands, only the band of low abundance with a molecular mass of 52 kDa (2) gave peptide fragments that were consistent with an in silico digest of the capsaicin synthase (Table 1). The size of 52 kDa also matched the predicted molecular mass of 49.2 kDa plus the 2 kDa His-tag that was detected by the antibody. As already illustrated by the SDS-PAGE pattern, the overall yields of the partially purified enzyme mix were low and never exceeded 0.3 mg L^−1^ of the *E. coli* culture, as calculated by the abundance of the capsaicin synthase band from SDS-PAGE. Band (1) in fraction A11 that eluted early in the SEC and showed the strongest UV_280nm_ signal after the SEC showed neither activity nor any reaction with the anti-His-tag antibody (Figure 2c,d). Nevertheless, this fraction was also analyzed by peptide sequencing since oligomerization of the capsaicin synthase to an inactive enzyme could not be excluded. However, this band corresponded to an *E. coli* heat shock protein (HSP) or 60 kDa chaperonine. The surprisingly high amount of this *E. coli* HSP60 complex in the IPTG-induced protein extracts transformed with the plasmid harboring the capsaicin synthase sequence is noteworthy. It might suggest that this complex is induced during expression and may be involved in the folding of an admittedly very limited amount of active recombinant capsaicin synthase. Protein bands (3) and (4) co-eluting with the capsaicin synthase were also identified as *E. coli* proteins.

### 2.2. Properties of Recombinant Capsaicin Synthase

Despite its very limited amounts, the partly purified capsaicin synthase could be used to determine some characteristics of the enzyme, including the substrate specificity and some kinetic data. The recombinant desalted capsaicin synthase was stable for a few weeks when stored at −80 °C at a concentration of 0.6 mg L^−1^ in a TRIS/HCl buffer of a pH 7.5–pH 8.0 containing 10% glycerol. When stored at 4 °C in the same buffer, the enzyme lost > 20% of its activity over 12 h. The enzyme activity was inhibited by imidazole and, therefore, desalting it into imidazole-free buffers was mandatory after the affinity purification of the His-tagged enzyme. The maximum activity was recorded in a slightly alkaline TRIS/HCl buffer consistent with reports of other BAHD-like enzymes, including cocaine synthase or piperine synthase [15,16]. An apparent temperature optimum for the partially purified enzyme preparation was determined at 35 °C where the enzyme was stable in the reaction buffer for at least 30 min (Appendix A). 

Except for the capsaicin, no additional, unspecific signals were observed when the capsaicin synthase was incubated with its substrates *trans* 8-methyl-6-nonenoyl-CoA and vanilloylamine. In order to test the capsaicin synthase for substrate specificity, in addition to *trans* 8-methyl-6-nonenoyl-CoA, a second CoA ester, 8-nonenoyl-CoA was synthesized from the corresponding acid and tested for enzyme activity. In addition, several commercially available aromatic and aliphatic CoA esters and a set of amines, but also alcohols, were tested (Figure 3; Table 2). 

Besides the natural substrate *trans* 8-methyl-6-nonenoyl-CoA, considerable product formation was only detected in the case of 8-nonenoyl-CoA and vanilloylamine. Minor product formation was also observed when hexanoyl-CoA was used as a substrate, whereas the tested aromatic CoA esters, benzoyl-CoA, coumaroyl-CoA, and feruloyl-CoA, could not be converted to an amide. Among the amines tested, only isobutylamine was converted to the corresponding 8-methyl-6-nonenoylisobutylamide and 8-nonenylisobutylamide, respectively, although at a much lower rate as compared to vanilloylamine. Additionally, no product formation was observed when vanilloylalcohol was used as a substrate instead of vanilloylamine. These data already indicate that capsaicin synthase is quite substrate-specific and consistent with the recently described piperine synthase that is specific for its substrates piperoyl-CoA and piperidine and also results in amide formation [16]. Other recombinant BAHD-like enzymes involved in plant ester formation are reportedly less specific [17]. 

Apparent kinetic constants were only determined for the natural substrate *trans* 8-methyl-6-nonenoyl-CoA and vanilloylamine. In the case of the CoA ester, the curve fits the classical Michaelis–Menten kinetics and the product yields dropped above 1 mM CoA substrate. In contrast, no matter how high the concentration of vanilloylamine (up to 25 mM in the assay), saturation with the substrate was never achieved. We also noticed that under the conditions used, a minimum concentration of 2 mM vanilloylamine was required for the recombinant enzyme reaction to proceed. Below this, even at longer incubation times, no product formation was observed at all.

Based on a single measurement with three technical replicates we, therefore, determined an apparent, calculated K_m_ of 0.48 mM ± 0.24 mM for the CoA-ester. In the case of vanilloylamine, this was not possible and might indicate a low affinity or a very high and saturating local substrate concentration in the capsaicin-producing cells (Appendix A). 

## 3. Discussion

In this investigation, we successfully established the enzyme activity of recombinant capsaicin synthase that catalyzes the final step in the biosynthesis of the pungent alkaloid, capsaicin. Although the *Pun1* locus of the *Capsicum* species was associated with pungency, efforts to produce an active protein up to now were unsuccessful. Even in the patent that claims the production of capsaicin in *E. coli* [12], any enzyme characteristics crucial for the biosynthesis of the pungent alkaloid capsaicin were not reported. Like most BAHD-type enzymes that catalyze the important steps in alkaloid formation, including cocaine synthase [15] and piperine synthase [16], capsaicin synthase was notoriously difficult to express and purify.

Structural information on capsaicin synthase is currently missing, although BAHD-like enzymes have been crystallized or modeled [18]. Besides the highly conserved HXXXD-motif required for the catalytic activity of this type of enzyme, capsaicin synthase shows only a 20% sequence identity, even to enzymes with a similar substrate preference, such as piperine synthase [16]. This makes it difficult to precisely predict individual amino acids or domains that affect substrate specificity. New neural network-based approaches, such as AlphaFold 2 [19], combined with established modeling tools, may provide structural insights into capsaicin synthase in the future. These predictions, besides the availability of CoA-activated acyl donors, always require experimental verification by fermentation [13] or, as in this case, enzyme assays with at least partly purified recombinant enzymes.

The specificity of capsaicin synthase could only be determined based on the synthesis of two CoA esters, the aliphatic *trans* 8-methyl-6-nonenoyl-CoA, the presumed precursor and, in parallel, the corresponding mono-unsaturated 8-nonenoyl-CoA from commercially available acids. Our data reveal that the enzyme is not absolute-specific for these precursors but may require a limited set of aliphatic CoA-donors, and also amine acceptors. Apparent K_m_ values for the substrates of capsaicin synthase are in the high three-digit micromolar range, in the same order of magnitude as the piperine synthase that produces a similar blend of amides [16] but more than 10-fold higher than the data reported for other recombinant BAHDs [14,15]. These surprisingly low affinities compared to other BAHD-type acyltransferases, specifically in the case of vanilloylamine, suggests that either high substrate concentrations are present in the cells where capsaicin biosynthesis takes place, as reported for piperine synthase [20], or that in vitro assay conditions are less optimal compared to the metabolic channeling proposed for specialized cells in vivo [21,22,23]. A rather high in vivo efficiency is consistent with the reported cell-specific accumulation of capsaicin during the rather short time frame of fruit development, resulting in concentrations of up to 10 g/kg of the dry weight of these alkaloids in the fruits of chili pepper [24].

Capsaicin biosynthesis is still far from being understood. A detailed investigation of the transcript accumulation of capsaicin biosynthesis revealed that some extremely pungent varieties, in addition to the septum, show a large expression of *Pun1* in the neighboring pericarp [23]. Interestingly, the study also reveals that some genes of the phenylpropanoid branch of the pathway, such as the gene encoding the aminotransferase (pAMT), which was proposed to catalyze the formation of the capsaicin synthase substrate, vanilloylamine, are not co-expressed with *Pun1* at the cellular level. These conflicts were also reported previously when a holistic model was developed for the capsaicin biosynthetic pathway [6]. Therefore, a more precise subcellular localization of individual pathway enzymes, such as capsaicin synthase and the proposed pAMT, is required to determine if the complete capsaicin formation takes place in the same cells or cell compartments, or if the shuffling of the precursors between different cell types is required, such as in the case of benzylisochinoline alkaloids in opium poppy or *Vinca*-alkaloids in *Catharanthus roseus* [25,26,27]. Since the stable transformation of chili peppers is still in its infancy [28], pure enzyme preparations may be required to obtain specific antibodies against capsaicin synthase and pAMT. In combination with laser microdissection, MS-imaging, and single-cell proteomics, this might reveal the precise subcellular localization of capsaicin biosynthesis in a highly relevant spice and prominent drug.

## 4. Materials and Methods

### 4.1. Substrate Synthesis

A standard of capsaicin was obtained from Merck (Darmstadt) and used as a reference to calculate enzyme activities and kinetic constants. Vanilloylamine, *trans* 8-methyl-6-nonenoyl carboxylic acid, 8-nonenoyl carboxylic acid, hexanoyl-CoA, and myristoyl-CoA were also obtained from Merck. Among several synthetic and enzymatic methods published for CoA-(bio)synthesis [29,30], the *trans* 8-methyl-6-nonenoyl-CoA and 8-nonenoyl-CoA esters used as substrates were synthesized, essentially, by established protocols using N-hydroxysuccinimide (NHS)/dicyclocarbodiimide (DCC) activation of the corresponding acids, as described in [31], and gave reliable results already in the case of piperoyl-CoA synthesis [32].

The precursor of capsaicin and in vivo substrate of capsaicin synthase, *trans* 8-methyl-6 nonenoyl-CoA ester, was prepared using the NHS/DCC activation strategy. *trans* 8-methyl-6-nonenoyl carboxylic acid (100 mg, 0.58 mmol) was added to a solution of NHS (66 mg, 0.58 mmol) in ethyl acetate (10 mL) at room temperature in a N_2_ atmosphere. A solution of DCC (120 mg, 0.58 mmol) in ethyl acetate (3 mL) was added dropwise. Stirring was continued for 12 h, after which TLC control (dichloromethane) showed complete conversion. The formed dicyclohexyl urea (DCU) precipitate was filtered off, the solvent was evaporated in a vacuum and subsequently eluted by chromatography on a silica gel (mesh 0.363–0.2 mm) with dichloromethane/methanol (98:2) and a retardation value (Rf) of 0.56 to yield 135 mg of a colorless solid. The product was analyzed by electrospray ionization-mass spectrometry (ESI-MS). Subsequently, the activated ester (85 mg, 296 µmol) in dioxane (8 mL) was added within 45 min to a solution of HS-CoA (93 mg, 118 µmol) in NaHCO_3_ (0.1 M, 4 mL) under a constant stream of N_2_. Stirring was continued for 2 h at room temperature. The TLC control (ALUGRAM RP-18W/UV_254_, Macherey-Nagel), ammonium acetate/methanol 52:48), showed complete conversion. To the reaction mixture was added ice water (20 mL), adjusted to a pH of 1.5–2 by 1N HCl, and extracted with ethyl acetate (4 × 30 mL). The aqueous phase was evaporated in a vacuum, upon which NH_4_OAc (10 mL, 0.01 M) was added. An HLB 20cc (1 g) LP Extraction Cartridge (Oasis, Waters, Eschborn) was pre-conditioned with 10 volumes of methanol and subsequently with 4% *w/v* NH_4_OAc. After loading, the column was washed with 4% *w/v* NH_4_OAc, and the product was eluted by a stepwise 10% increase of 100% MeOH in water. Purification was monitored by TLC (ALUGRAM RP-18/UV_254_) with a solvent mixture of aqueous 10 mM NH_4_OAc:MeOH 52:48. The thioester-containing fractions were concentrated by rotary evaporation (30 °C) and lyophilized to yield a colorless solid (104 mg). The product was stable for several weeks when stored at –80 °C, either dry or dissolved in 25% DMSO. 

TLC (ALUGRAM RP-18W/UV_254_): 0.28, NH_4_OAc:MeOH 52:48; ^1^H NMR (400 MHz, MeOH-d4; Varian Mercury 400 NMR spectrometer) δ 8.56 (d, *J* = 8.5 Hz, 1H, C*H*), 8.18 (d, *J* = 5.4 Hz, 1H, C*H*), 6.12 (d, *J* = 6.0 Hz, 1H, 1-*H*), 5.79 (ddt, *J* = 17.0, 10.2, 6.7 Hz, 1H, 2-*H*), 4.81 (t, *J* = 5.5 Hz, 1H), 4.52–4.46 (m, 1H, C*H*), 4.33–4.18 (m, 2H, 35-*H*), 4.05 (d, *J* = 4.8 Hz, 1H, 21-*H*), 4.00 (dd, *J* = 9.7, 5.2 Hz, 1H, C*H*), 3.58 (dd, *J* = 9.7, 4.2 Hz, 1H, C*H*), 3.46 (td, *J* = 6.8, 2.9 Hz, 2H, C*H*_2_), 3.29 (d, *J* = 5.9 Hz, 6H, C*H* + C*H*_2_), 2.98 (td, *J* = 6.9, 2.8 Hz, 2H, C*H*_2_), 2.56 (td, *J* = 7.4, 2.4 Hz, 2H, C*H*_2_), 2.42 (t, *J* = 6.8 Hz, 2H, C*H*_2_), 2.08–1.98 (m, 2H, C*H*_2_), 1.62 (tt, *J* = 9.2, 4.6 Hz, 2H, C*H*_2_), 1.43–1.26 (m, 6H, 2 × C*H*_3_), 1.05 (s, 3H, C*H*_3_), 0.84 (d, *J* = 4.4 Hz, 3H, C*H*_3_).; ^13^C NMR (101 MHz, Methanol-d4; Varian Mercury 400 NMR spectrometer) δ 200.71 (C-9), 178.55, 175.66, 173.91, 157.25 (C-51), 153.86 (C-53), 153.78 (C-50), 141.08 (C-47), 139.98 (C-2), 120.17 (C-49), 114.83 (C-1), 88.89 (CH), 84.58 (d, CH), 75.76 (d, CH), 75.46 (d, CH), 75.08 (C-21), 73.28 (d, CH), 66.27 (t, CH), 49.85, 44.78, 40.12 (t, C-23), 36.55, 36.38, 34.75, 29.83, 29.76, 29.08, 26.61, 22.98 (CH_3_), 22.22 (CH_3_), 19.59 (CH_3_), 19.51 (CH_3_).; NMR-spectra of the newly synthesized CoA-ester are displayed in Appendix A.; ESI-MS (API 3200, AB Sciex) C_31_H_48_Li_4_N_7_O_17_P_3_S calc. 943.50, Appendix A).8-nonenoyl-CoA ester was also synthesized by NHS/DCC activation. 8-Nonenoyl carboxylic acid (0.66 g, 4.2 mmol) was added to a solution of NHS (0.48 g, 4.2 mmol) in ethyl acetate (20 mL) at room temperature in a N_2_ atmosphere. A solution of DCC (0.86 g, 4.22 mmol) in ethyl acetate (6 mL) was added dropwise. Stirring was continued for 12 h, after which the TLC control showed complete conversion. The formed dicyclohexylurea (DCU) precipitate was filtered off, the solvent was evaporated in a vacuum, and subsequently eluted by chromatography on a silica gel (0.363–0.2 mm) with dichloromethane and Rf-values 0.85 in dichloromethane/methanol 100:2) to yield 673 mg of a colorless solid. The product was analyzed by ESI-MS. The ester (33 mg, 0.13 mmol) in dioxane (6 mL) was added within 45 min to a solution of HS-CoA (40 mg, 0.05 mmol) in NaHCO_3_ (0.1 M, 4 mL)under a constant stream of N_2_. Stirring was continued for 2 h at room temperature. The TLC control (ALUGRAM RP-18W/UV_254_, Macherey-Nagel), ammonium acetate/methanol 52:48), showed complete conversion. To the reaction mixture was added ice water (20 mL), adjusted to a pH of 1.5–2 by 1N HCl, and extracted with ethyl acetate (4 × 30 mL). The aqueous phase was evaporated in a vacuum, upon which NH_4_OAc (10 mL, 0.01 M) was added. An HLB 20cc (1 g) LP Extraction Cartridge (Oasis) was pre-conditioned with 10 volumes of methanol and subsequently with 4% *w/v* NH_4_OAc. After loading, the column was washed with 4% *w/v* NH_4_OAc, and the product was eluted by a stepwise 10% increase of 100% MeOH in water. Purification was monitored by TLC (ALUGRAM RP-18/UV_254_) with a solvent mixture of aqueous 10 mM NH_4_OAc:MeOH 52:48. The thioester-containing fractions were concentrated by rotary evaporation (30 °C) and lyophilized to yield a colorless solid (29.3 mg). The product was stable for several weeks when stored at −80 °C, either dry or dissolved in 25% DMSO. 

TLC (ALUGRAM RP-18W/UV_254_): 0.28, NH_4_OAc:MeOH 52:48; ^1^H NMR (400 MHz, Methanol-*d*_4_; Varian Mercury 400 NMR spectrometer) δ 8.60–8.52 (s, 1H, C*H*), 8.22–8.16 (s, 1H, C*H*), 6.15–6.09 (d, *J* = 5.9 Hz, 1H, C*H*), 5.43–5.11 (m, 2H, 3-H + 4-*H*), 4.84–4.77 (t, *J* = 5.5 Hz, 1H), 4.51–4.44 (p, *J* = 3.1 Hz, 1H, C*H*), 4.32–4.19 (m, 2H, *H*-35), 4.08–4.04 (s, 1H, 21-*H*), 4.04–3.96 (dd, *J* = 9.7, 5.2 Hz, 1H, C*H*), 3.61–3.53 (dd, *J* = 9.7, 4.1 Hz, 1H, C*H*), 3.52–3.40 (td, *J* = 6.9, 3.5 Hz, 2H, C*H*_2_), 3.31–3.24 (m, 3H, C*H* + C*H*_2_), 3.02–2.94 (t, *J* = 6.8 Hz, 2H, C*H*_2_), 2.61–2.52 (td, *J* = 7.4, 3.1 Hz, 2H, C*H*_2_), 2.45–2.37 (t, *J* = 6.8 Hz, 2H, C*H*_2_), 2.29–2.15 (dp, *J* = 13.1, 6.7 Hz, 1H, 2-*H*), 2.08–1.96 (m, 1H), 1.99–1.93 (m, 1H), 1.68–1.56 (p, *J* = 7.5 Hz, 2H, C*H*_2_), 1.42–1.25 (m, 2H, C*H*_2_), 1.08–1.04 (s, 3H, C*H*_3_), 0.98–0.90 (m, 6H, 1, 2 × C*H*_3_), 0.85–0.81 (s, 3H, C*H*_3_); ^13^C NMR (101 MHz, Methanol-*d*_4_; Varian Mercury 400 NMR spectrometer) δ 200.63 (*C*-9), 179.39, 175.66, 173.87, 157.22 (*C*-51), 153.84 (*C*-53), 150.93 (*C*-50), 141.05 (*C*-47), 139.26 (*C*-3), 127.66 (*C*-4), 120.15 (*C*H), 88.88 (*C*H), 84.6(d, *C*H), 75.77 (d, *C*H), 75.25 (d, *C*H), 75.02 (*C*H), 73.28 (d, *C*H_2_), 66.29 (t, *C*H), 44.64, 40.10 (t, *C*-23), 36.54, 36.37, 33.10, 32.26 (*C*H), 30.10, 29.86, 29.07, 27.82, 27.55, 26.22, 26.07, 23.58 (*C*H_3_), 23.11 (*C*H_3_), 22.25 (*C*H_3_), 19.42 (*C*H_3_); NMR-spectra of the newly synthesized CoA-ester are displayed in Appendix A; ESI-MS (API 3200, AB Sciex) C_31_H_48_Li_4_N_7_O_17_P_3_S calc. 943.50 (Appendix A).

### 4.2. Cloning and Functional Expression of Recombinant Capsaicin Synthase

The codon-optimized sequence of capsaicin synthase (Stewart et al., 2005) was synthesized by GeneArt (Ebersberg, Germany), digested with NdeI and XhoI, and resulted in a fragment of 1332 bp that was subsequently cloned into the vector pET-16b (Merck, Darmstadt, Germany) and transformed into electrocompetent *E. coli* SoluBL21 cells (Amsbiotec, Abingdon, UK). A series of colonies were grown in 5 mL of LB and each was supplemented with 100 mg L^−1^ ampicillin (LB_amp_) to an OD_600_ of 0.4, induced with 1 mM IPTG, grown for 15 h at 25 °C, and lysed and checked for enzyme activity. A single positive clone was then grown in 200 mL of LB_amp_ as the selectable marker under constant shaking (170 rpm) at 25 °C, added to 1.5 L LB_amp_, to reach an OD_600_ of 0.2, incubated at 25 °C at 120 rpm to an OD_600_ 0.4, induced by 1 mM IPTG, and subsequently grown for 21 h. Cells were harvested by centrifugation for 20 min at 8000 g, the supernatant was removed, and the pellet was suspended in 200 mL of H_2_O and centrifuged at 10,000 g for 15 min. Cells were re-suspended in 20 mL of a protein buffer (50 mM TRIS/HCl pH 7.5, 150 mM NaCl, 1 mM TCEP (Tris(2-carboxyethyl)phoshine), 10% glycerol) and protease inhibitor (Roche). Lysis was performed by adding 1 mg L^−1^ of lysozyme, followed by French Press disruption at 4 °C. After centrifugation at 4 °C for 10 min at 15,000 g, the supernatant was stirred for 5 min at room temperature with 0.05% protamine sulfate in H_2_O to remove nucleic acids and centrifuged again at 4 °C for 20 min at 15,000 g. The supernatant was then applied to a 5 mL Ni-NTA affinity column in 20 mM of TRIS/HCl at a pH of 7.5 (Macherey-Nagel, Düren, Germany) and His-tagged capsaicin synthase was eluted with 300 mM imidazole in 20 mM of TRIS/HCl at a pH of 7.5. Enzymatically active fractions (for details, see Chapter 4.4) were combined, desalted into a protein buffer, concentrated on a Centricon 30 membrane device (Merck, Darmstadt, Germany), and fractionated by size exclusion chromatography on a Superdex Increase 10/300 (Cytiva, Freiburg, Germany) at a flow rate of 0.4 mL min^−1^ with a buffer of 50 mM of TRIS/HCl at a pH of 7.5, 5% glycerol, and 150 mM of NaCl. Active fractions tested by LC-MS were combined and used for the determination of the enzyme properties, substrate specificity, as well as kinetic data. Protein purification was checked on 10% SDS-PAGE gels and proteins stained by Coomassie Brilliant Blue G 250 (Serva, Heidelberg, Germany). For verification of the correct protein band, three purified bands of an appropriate size and elution profile, matching the activity profile and, therefore, potentially corresponding to active capsaicin synthase, were cut out, subjected to peptide sequencing, and blasted against the SWISS-PROT database.

### 4.3. Protein Digestion and LC-MS/MS-Based Analysis

SDS-PAGE separated protein bands 1–4 were in-gel digested with trypsin and desalted, as described in a previous report [33]. Dried peptides were dissolved in 5% acetonitrile and 0.1% trifluoroacetic acid and injected into an EASY-nLC 1000 liquid chromatography system (Thermo Fisher Scientific, Waltham, MA, USA). Peptides were separated using liquid chromatography C_18_ reverse phase chemistry, employing a 120 min gradient increasing from 5% to 40% acetonitrile in 0.1% formic acid and a flow rate of 250 nL min^−1^. Eluted peptides were electrosprayed on-line into a QExactive Plus mass spectrometer (Thermo Fisher Scientific). The spray voltage was 1.9 kV, the capillary temperature was 275 °C, and the Z-Lens voltage was 240 V. A full MS survey scan was carried out with the chromatographic peak width set to 15 s, the resolution set to 70,000, an automatic gain control (AGC) of 3 × 10^6^, and a max injection time (IT) of 100 ms. MS/MS peptide sequencing was performed using a Top10 DDA scan strategy with HCD fragmentation. MS scans with mass-to-charge ratios (*m/z*) between 400 and 1850 were acquired. MS/MS scans were acquired with a resolution of 17,500, an AGC of 5 × 10^4^, an IT of 50 ms, an isolation width of 1.6 *m/z*, normalized collision energy of 28, an underfill ratio of 3%, a dynamic exclusion duration of 20 s, and an intensity threshold of 3 × 10^4^.

Peptides and proteins were identified using the Mascot software v2.7.0 (Matrix Science) linked to Proteome Discoverer v 2.1 (Thermo Fisher Scientific). A precursor ion mass error of 10 ppm and a fragment ion mass error of 0.02 Da were tolerated in searches of the *Arabidopsis thaliana* TAIR10 database amended with common contaminants and capsaicin synthase (pun1-gen) proteins. Carbamidomethylation of cysteine (C) was set as a fixed modification and the oxidation of methionine (M) was tolerated as a variable modification. A spectrum (PSM), peptide, and protein level false discovery rate (FDR) was generated for all annotated PSMs, the peptide groups, and proteins based on the target-decoy database model, and the Target Decoy PSM Validator module. PSMs, peptide groups, and proteins with q-values beneath the significance threshold of α = 0.01 for the PSMs and peptide groups and 0.05 for the proteins were considered identified.

### 4.4. Enzyme Assays and Product Analytics

For enzyme assays, 3 µg of the partly purified recombinant enzyme were incubated with 1 mM of activated CoA-ester (kinetics 0.05 mM up to 4 mM), 2 mM of vanilloylamine (kinetics 100 µM up to 20 mM), 50 mM of TRIS/HCl at a pH of 7.5, 150 mM of NaCl, 10% glycerol, and 1 mM of TCEP in a final volume of 50 µL for up to 30 min. TCEP was included in all buffers to prevent the formation of inactive oligomers. A reaction mixture without an enzyme to detect and monitor spontaneous capsaicin formation was incubated for the same time and was used as an additional control. Reactions were stopped by the addition of 10 µL of a mixture of 50% acetonitrile/10% formic acid (*v/v*), kept on ice for 5 min, centrifuged to precipitate the protein, and analyzed by reversed-phase HPLC. Capsaicin formation was monitored on a 50/3 mm C_18_ reverse phase Nucleosil column (Macherey-Nagel) at a flow rate of 0.4 mL min^−1^ and a gradient from a 70% aqueous 0.1% formic acid (solvent A) and 30% acetonitrile (solvent B) to 80% solvent B within 7.5 min. Products were analyzed on an e2695 chromatography workstation equipped with a photodiode array detector (PDA) and a Auquity^®^ QDA-mass detector (Waters, Eschborn, Germany). Products were recorded simultaneously by UV/Vis detection between 240–400 nm (if applicable). ESI-MS mass detection of capsaicin at *m/z* 306 [M + H]^+^ was performed in the positive ionization mode and between *m/z* 200–400 [M + H]^+^. The cone voltage was set at 15 V. All reactions, standard curves, and assays were run in technical triplicates, and the average value was recorded. 

## 5. Conclusions

In summary, based on the chemical synthesis of substrate CoA esters and the expression of recombinant capsaicin synthase in *E. coli*, we have successfully characterized capsaicin synthase activity. Although based on genetic evidence, its role in capsaicin biosynthesis has never been questioned. The functional characterization of capsaicin synthase was elusive and, therefore, the enzyme characteristics remained enigmatic. From our data, it is evident that the enzyme does not require any co-factors or specific posttranslational modifications. The specificity towards aliphatic as compared to aromatic CoA esters is noteworthy. Only a limited set of amines and no alcohols are accepted as substrates. A combination of modeling, substrate docking, and site-directed mutagenesis can now be used to design and engineer capsaicin synthase activities for optimized catalytic properties and desired substrate preferences. 

## Figures and Tables

**Figure 1 molecules-27-06878-f001:**
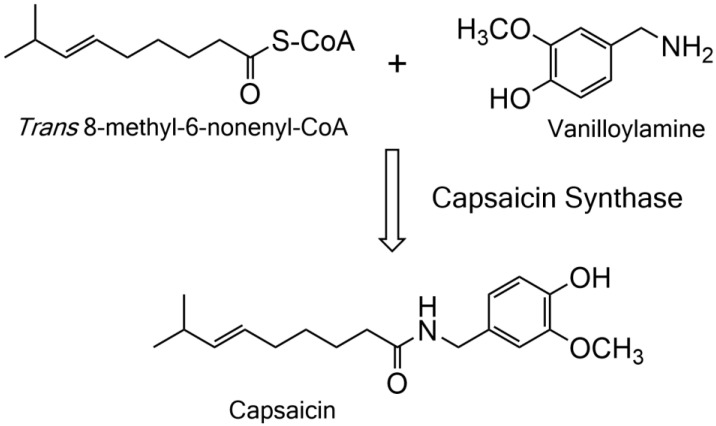
Reaction catalyzed by capsaicin synthase in chili peppers. An amide bond is formed between *trans* 8-methyl-6-nonenoyl-CoA and vanilloylamine by capsaicin synthase, a BAHD-type acyltransferase.

**Figure 2 molecules-27-06878-f002:**
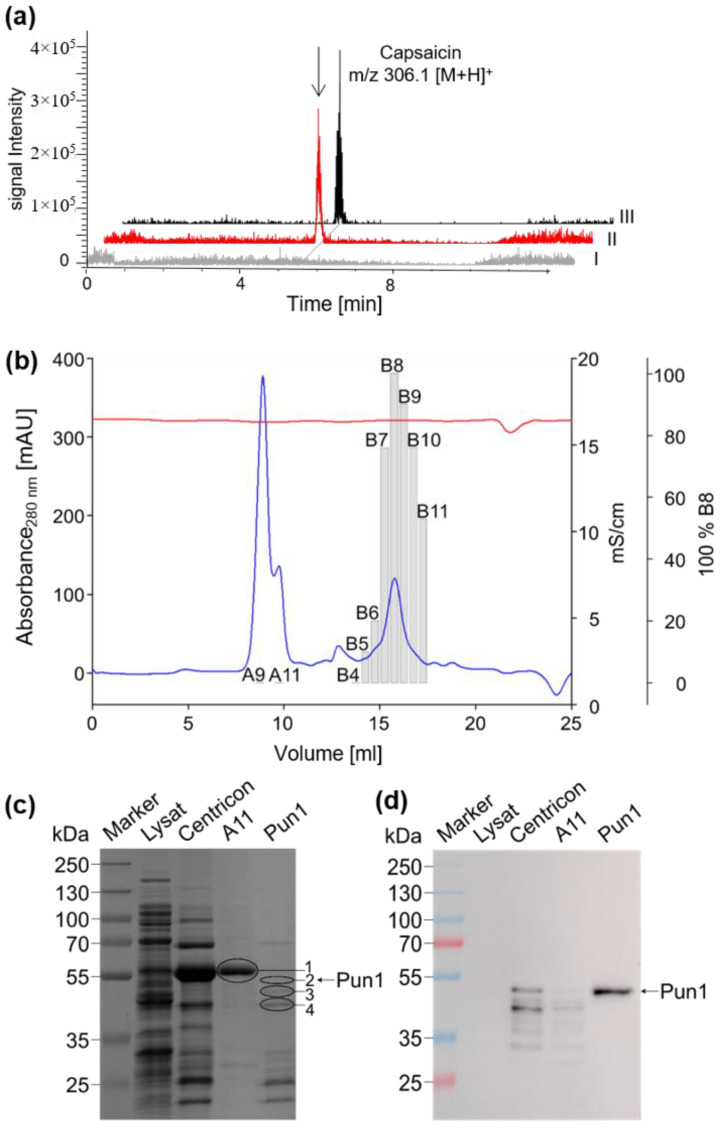
(**a**) Partial purification of recombinant capsaicin synthase. Capsaicin product formation (II) was detected in crude enzyme extracts. Retention time and mass signals were identical to a commercially available standard of capsaicin (III), *m/z* 306.1 [M + H]^+^. Concentrated capsaicin synthase was purified by size exclusion chromatography, (**b**) and the UV_280nm_ and activity signals are overlaid. Gray bars represent the relative enzyme activity of individual fractions as compared to fraction B8, which was set to 100%. (**c**) and (**d**) represent the SDS-PAGE protein profile and the corresponding Western blot of individual purification steps. The three dominant bands corresponding to the combined SEC fractions, B7-B10, shown in (**b**) at 52 kDa (2), 50 kDa (3), and 48 kDa (4), were cut out from the SDS-PAGE gel together with band (1), correlating to the strongest UV_280nm_ signal in (**b**) but without any activity. All were checked for the presence of capsaicin synthase by tryptic digest and subsequent peptide sequencing.

**Figure 3 molecules-27-06878-f003:**
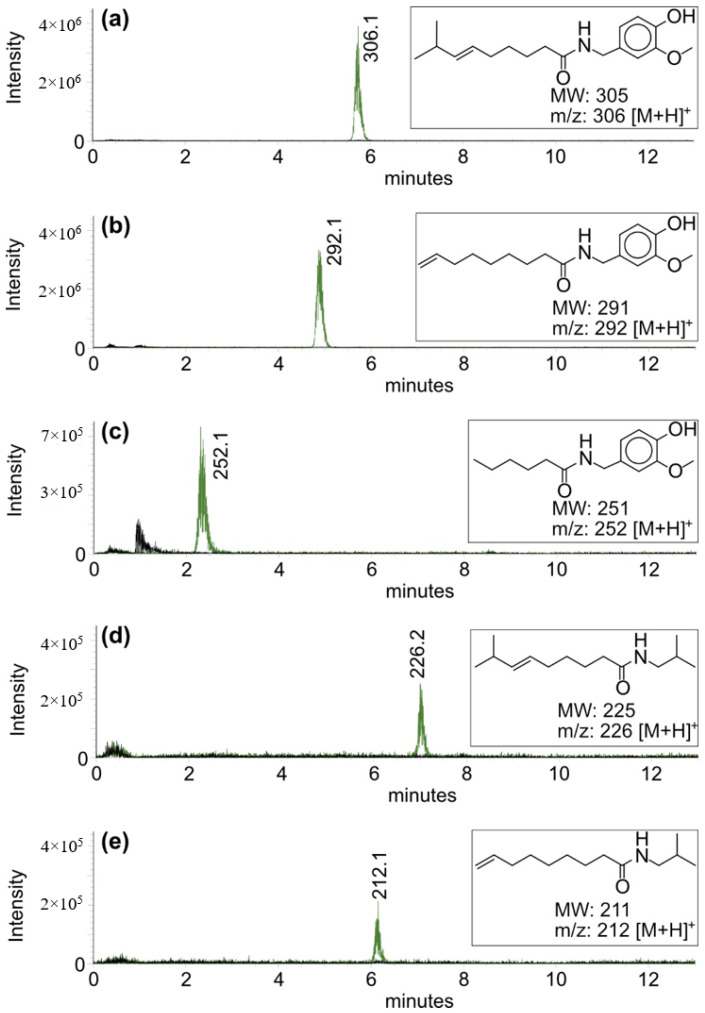
Product formation of recombinant capsaicin synthase with a set of aliphatic CoA-esters. Formation of (**a**), capsaicin; (**b**), 8-nonenoyl-vanilloylamide; (**c**), hexanoyl-vanilloylamide; (**d**), *trans* 8-methyl-6-nonenoyl-isobutylamide; (**e**), 8-nonenoyl-isobutylamide.

**Table 1 molecules-27-06878-t001:** Identification of capsaicin synthase purified by SDS-PAGE. Tryptic digest of band 2 (Figure 2) and sequence comparison against the SwissProt database revealed the highest mascot scores and highest peptide coverage to the proposed *Pun 1*-encoded acyltransferases of *C. frutescens*, *C. chinense*, and *C. annuum.* Fragments of trypsin were the second most abundant peptides identified from the tryptic digest of this protein band.

Accession/Organism	Annotation	Peptide Score	Total Peptides	Calculated Mass [kDa]/pI	Total Mascot Score
Q58VT1 *C. frutescens*	Acyltransferase *Pun1*	25.868	9	49.3/6.95	837
A0A2G3D6U1 *C. chinense*	Acyltransferase *Pun1*	22.082	8	49.2/6.77	769
D2Y3X2 *C. annuum*	Acyltransferase *Pun1*	11.364	5	49.3/7.72	492
P00761	Trypsin	6.894	3	24.4/7.18	207

**Table 2 molecules-27-06878-t002:** Matrix of the substrate specificity of recombinant capsaicin synthase. The activity of the partially purified enzyme was calculated to the 240 nkat mg^−1^ protein and set to 100% (measured as the average of 3 technical replicates). Relative values are calculated in the case of other substrates (n.d. = product formation not detectable; (−) = not tested).

Substrate Combination	Agmatine	Isobutylamine	Vanilloylalcohol	Vanilloylamine
8-Methyl-6-nonenoyl-CoA	n.d.	6	n.d.	100
8-Nonenoyl-CoA	n.d.	4	n.d.	112
Hexanoyl-CoA	n.d.	n.d.	n.d.	22
Myristoyl-CoA	n.d.	n.d.	n.d.	n.d.
4-Coumaroyl-CoA	(−)	n.d.	(−)	n.d.
Feruolyl-CoA	(−)	n.d.	(−)	n.d.
Benzoyl-CoA	(−)	n.d.	(−)	n.d.

## Data Availability

All raw data have been transferred to the open data depository www.radar-service.eu and will be publicly available at https://doi.org/10.22000/799. Clones and sequences as well as LC/MS files and NMR data are also stored on local servers and hard drives and are available upon reasonable request from the senior authors BW and TV.

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
