# Peer review of "Chemical Synthesis of *Trans* 8-Methyl-6-Nonenoyl-CoA and Functional Expression Unravel Capsaicin Synthase Activity Encoded by the *Pun1* Locus"

_molecules, 2022, doi:10.3390/molecules27206878_

Round 1
Reviewer 1 Report
The work is well done and the manuscript is well written and deserves publication.
Author Response
Thank you for your positive comments.
Reviewer 2 Report
The authors have characterized in vitro the acetyl transferase activity of Pun1 involved in the biosynthesis of capsaicinoids. They have shown this enzyme does not need co-factors, has a narrow substrate specificity and is only saturable by the acyl chain substrate (Km 0.48mM) and not by the amine substrate (not saturable at 25 mM).
The article summarizes a very through approach to lift an important hurdle in our comprehension of capsaicinoid synthesis. I have no remarks other than some grammatical erros such as line 57 use rather that than this.
Author Response
Thank you very much for your encouraging comments. We modified the corresponding "this" and also rechecked the manuscript for grammatical errors ccording to your suggestion.
Reviewer 3 Report
The manuscript is clearly written. I have some comments to the paragraph about protein purification:
Paragraph 4.2.:
line 333: abbreviation LBamp is used, description is in line 335, but its nopt clear whether the conc of ampicin was the same.
line 333: "a series of 5 mL colonies" is I guess a short name for a process of picking a colony and grow in 5 mL. Could be clarified.
line 336: "selectable marker" should be replaced for Selection marker.
Whats the retional for washing and entrifuging the pellet first in water?
line 348: active fractions - was the activity of each fraction checked with the procedure described in 4.4?
General:
What is the structural knowledge about capsaicin synthase?
Author Response
Reviewer comments: The manuscript is clearly written. I have some comments to the paragraph about protein purification:
Answer: Thank you very much for your encouraging statement and the additional suggestions that we have addressed in detail as follows:
Reviewer comments: Paragraph 4.2.: line 333: abbreviation LBamp is used, description is in line 335, but its nopt clear whether the conc of ampicin was the same; line 333: "a series of 5 mL colonies" is I guess a short name for a process of picking a colony and grow in 5 mL. Could be clarified and line 336: "selectable marker" should be replaced for Selection marker.
Answer: You are absolutely correct, we modified these somewhat sloppy and potentially misleading statements in this chapter according to your suggestion.
Reviewer comment: Whats the retional for washing and entrifuging the pellet first in water?
Answer: By recentrifugation in water, we remove residual impurities and the rest of the growth medium and gain a pellet that is less viscous for the subsequent purification step on the metal affinity purification on NiNTA prepacked columns in principle, it elutes faster. You can do without it, but the additional water wash before freezing (you may also use a buffer of low ionic strength) reduces the exposure time of this fragile enzyme in the crude protein extract to potential protease degradation.
Reviewer comment: line 348: active fractions - was the activity of each fraction checked with the procedure described in 4.4?
Answer: Thank you very much for this hint, yes, in all cases enzyme and substrate concentrations were used as described in 4.4. We added a short statement in teh text to check chapter 4.4 for details.
Reviewer comment: General: What is the structural knowledge about capsaicin synthase?
Answer: Thank you very much for this question. It is an important point. At this stage it was not our intention to work on the unknown structural details of the enzyme. Currently, crystallization seems not possible, due to low yield and instability of the enzyme. A combination of Alpha-fold prediction and modeling may come up with catalytically relevant amino acids beyond the conserved HXXXD motif that is required for the catalytic mechanism. However, all BAHDs share limited sequence identities and very few conserved regions. E.g. although capsaicin synthase and piperine synthase (that we purified recently) share similar substrates, sequence identity is only 20 %. We added a few statements including two references in the discussion.